

# Complementarity of Wind Measurements from Co-located X-band Weather Radar and Doppler Lidar

Jenna Ritvanen[1,2], Ewan O'Connor[1], Dmitri Moisseev[1,2], Raisa Lehtinen[3], Jani Tyynelä[1], and Ludovic Thobois[4]

[1]Finnish Meteorological Institute, Helsinki, Finland
[2]Institute for Atmospheric and Earth System Research, Faculty of Science, University of Helsinki, Helsinki, Finland
[3]Vaisala Oyj, Vantaa, Finland
[4]Vaisala France SAS, Paris-Saclay, France

**Correspondence:** Jenna Ritvanen (jenna.ritvanen@fmi.fi)

**Abstract.** Accurate wind profile measurements are important for applications ranging from aviation to numerical weather prediction. The spatial pattern of winds can be obtained with ground-based remote sensing instruments, such as weather radars and Doppler lidars. As the return signal in weather radars is mostly due to hydrometeors or insects, and in Doppler lidars due to aerosols, the instruments provide wind measurements in different weather conditions. However, the effect of various weather
conditions on the measurement capabilities of these instruments has not been previously extensively quantified. Here we present results from a 7-month measurement campaign that took place in Vantaa, Finland, where a co-located Vaisala WRS400 X-band weather radar and WindCube400S Doppler lidar were employed continuously to perform wind measurements. Both instruments measured plan-position-indicator (PPI) scans at 2.0 degrees elevation from horizontal. Direct comparison of radial Doppler velocities from both instruments showed good agreement with $R^2 = 0.96$. We then examined the effect of horizon-
tal visibility, cloud base height, and precipitation intensity on the measurement availability of each instrument. The Doppler lidar displayed good availability in clear air situations and the X-band radar in precipitation. Both instruments exhibited high availability in clear air conditions in summer when insects were present. The complementary performance in measurement availability of the two instruments means that their combination substantially increases the spatial coverage of wind observations across a wide range of weather conditions.

## 1 Introduction

Accurate wind profile measurements are required by numerous applications, ranging from assimilation in numerical weather prediction (NWP) models to monitoring weather conditions in weather-sensitive industries, such as aviation. The World Meteorological Organization (WMO) Statement of Guidance for Global NWP states that, "wind profiles at all levels outside the main



populated areas are a top priority among variables that are not adequately measured by current or planned systems" (Andersson, 2017).

Remote sensing instruments, such as Doppler lidars and Doppler weather radars, are appealing technologies for measuring wind, since they can provide wide spatial coverage of near-surface wind measurements. Doppler lidars measure the radial wind velocity based on the Doppler effect on the signal backscattered from aerosols (Werner, 2005). Given a suitable concentration

of atmospheric aerosols, Doppler lidars can measure the radial wind velocity out to ranges of 10 to 20 km depending on the instrument. However, strong attenuation of the lidar beam by precipitation, fog and cloud will often limit the measurement availability to much closer ranges (Guo et al., 2015; Liu et al., 2019).

Operating at longer wavelengths than Doppler lidar, Doppler weather radars measure the radial velocity from scattering by hydrometeors (Rauber and Nesbitt, 2018). However, their measurement availability is often limited in clear air conditions.

Biological scatterers, such as insects in the boundary layer during the warm season, can provide radial velocity measurements in clear air conditions, as can Bragg scattering, caused by fluctuations in air refractive index, which can be observed at S- and C-band (Achtemeier, 1991; Wilson et al., 1994; Contreras and Frasier, 2008; Franck et al., 2021). Operating at higher frequencies, such as X-band, offers several advantages over S- and C-band (McLaughlin et al., 2009), including ease of co-location with Doppler lidar in critical environments such as airports.

The measurement limitations suggest that, for all-weather capability, neither instrument is sufficient on its own as there will be significant gaps in data availability for particular situations (Nijhuis et al., 2018). However, with different scattering media responsible for Doppler lidar and weather radar observations, the combined data availability from both instruments may close the gaps in data availability significantly. Therefore, we quantify the performance of both instruments in various weather conditions in order to demonstrate the complementarity of radar and lidar wind observations. The study uses observations

collected during a 7-month measurement campaign in Vantaa, Finland where a co-located Vaisala WindCube400S Doppler lidar and Vaisala WRS400 X-band weather radar were deployed to perform simultaneous wind measurements. These observations were supplemented by surface station measurements of horizontal visibility, cloud base height, and precipitation intensity, in order to characterise the weather conditions.

The rest of this article is structured as follows. Section 2 describes the measurement campaign and instruments. The analysis

methods are described in Section 3, and the results comparing the Doppler lidar and X-band weather radar Doppler velocity measurements and their availability are presented in Section 4. Finally, Section 5 summarises the main findings.

## 2 Data

### 2.1 Measurement campaign and site

A Vaisala WindCube400S Doppler lidar (Dolfi-Bouteyre et al., 2008; Thobois et al., 2019) and a Vaisala WRS400 X-band

weather radar were deployed a few metres apart (see Fig. 1) on the roof of a building at Vaisala Oyj headquarters in Vantaa, Finland (60°16'56.6" N 24°52'33.9" E, 55 m above mean sea level). Both instruments were configured to measure plan-position-indicator (PPI) scans with an elevation angle of 2.0° from horizontal. The campaign period was from 1 May to 30



November 2021, however the WindCube400S lidar was out of operation due to a broken component from 22 June to 17 August. The lidar and X-band radar measurements are described in detail in Section 2.2 and 2.3, respectively.

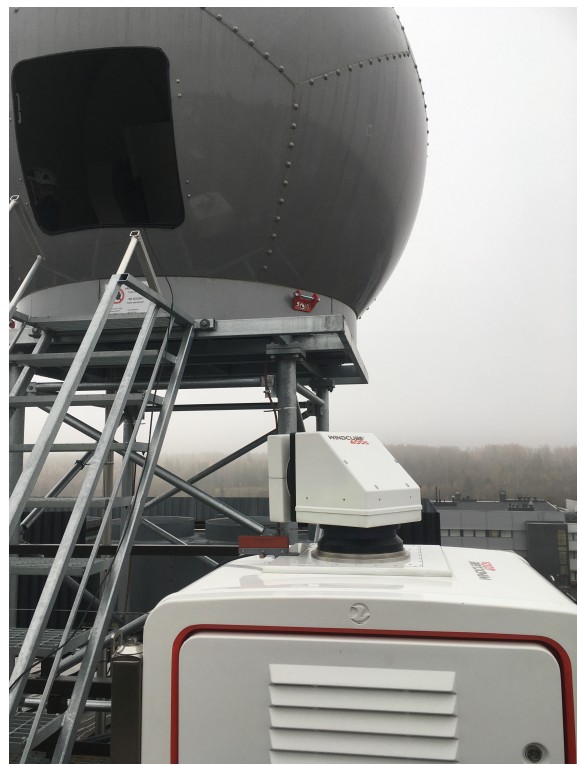

**Figure 1.** Vaisala WRS400 X-band radar and WindCube400S Doppler lidar installed on the roof of a building at Vaisala Oyj headquarters in Vantaa, Finland. Photo taken by Raisa Lehtinen.

The measurement location was chosen for convenience reasons; however, it is not optimal. For both instruments, nearby trees blocked specific azimuths and, additionally, for the lidar, a sector to the east was blocked by the X-band radar. The blocked rays are clearly seen in Fig. 2, which presents the fraction of available 2.0° PPI measurements during the entire measurement campaign for each instrument. For analysis purposes, any ray with a measurement availability of less than 5% was considered to be blocked.

Since the instruments were located at the same location and altitude, and the analysis presented here is limited to be within the lidar measurement range (i.e. within 14.3 km), we considered the lidar beam and the centre of the X-band radar beam to follow the same path. Additionally, the lidar and X-band radar measurements were temporally synchronised so that the scan starting times were as close as possible. Therefore, this measurement campaign provided a unique opportunity to compare Doppler velocity measurements from both instruments.





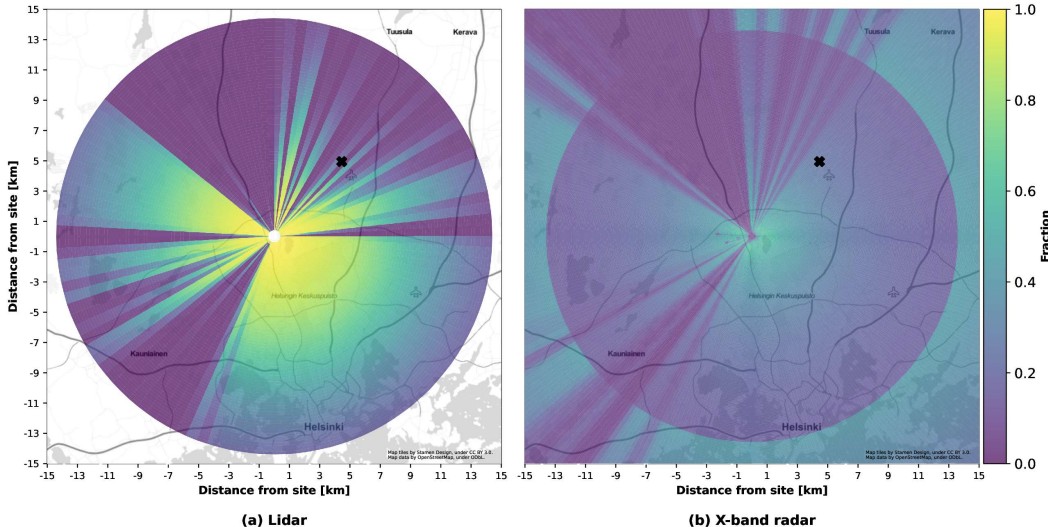

**Figure 2.** Fraction of available measurements during the entire measurement campaign for (a) lidar and (b) X-band radar. The location of the surface weather station is marked with a black cross. Background map tiles by Stamen Design, under CC BY 3.0. Map data by OpenStreetMap, under ODbL.

## 2.2 Doppler wind lidar

The Vaisala WindCube400S Doppler wind lidar is a coherent pulsed Doppler lidar operating at a wavelength of 1.54 μm (Dolfi-Bouteyre et al., 2008). Coherent Doppler lidars typically use high pulse repetition rates and accumulate many pulses in order to achieve the desired sensitivity. The high pulse repetition rate limits the maximum range of the system, however, this is not a major issue since signals close to the maximum range are normally far below the detection threshold. Radial velocities are obtained via the heterodyne technique (Frehlich and Kavaya, 1991) and the maximum unambiguous velocity depends on the sampling rate. The lidar beam itself is very narrow and, when scanning, the angular resolution of a ray is a function of the pulse accumulation time.

The instrument-provided data was ingested and processed using the open source package wradlib (Heistermann et al., 2013) and the radial velocity data then filtered using the quality flags provided by the instrument and a Carrier-to-Noise Ratio (CNR) threshold of -30 dB. No clutter filtering was applied to the Doppler lidar data.

Other relevant lidar specifications are summarised in Table 1.

## 2.3 Weather radar

The Vaisala WRS400 X-band weather radar uses solid state transmitters for horizontal and vertical polarisation channels. To achieve the required sensitivity, a 44 μs pulse with frequency modulation and corresponding pulse compression is used (Mudukutore et al., 1998; Bharadwaj and Chandrasekar, 2012; O'Hora and Bech, 2007). Because the use of such a long pulse





**Table 1.** Vaisala WindCube400S Doppler wind lidar specifications

| **Lidar parameter** | |
| --- | --- |
| Lidar wavelength | 1.54 μm |
| Laser divergence | 33 μrad |
| Pulse duration | 800 ns |
| Pulse repetition rate | 10 kHz |
| Max. unambiguous velocity | 30.4 m s$^{-1}$ |
| Range resolution | 200 m |
| Maximum range | 14.3 km |
| Number of pulses | 1000 |
| Elevation angle | 2.0° |
| Ray angle resolution | 3.0° |

leads to a blind region in the vicinity of the radar, a shorter pulse of 1 μs is used in addition (Bharadwaj and Chandrasekar, 2012). The transition from the short pulse to the long pulse measurements occurs at 13.5 km.

The Doppler observations are performed using a dual pulse repetition frequency (dual-PRF) pulsing scheme with pulse repetition frequencies (PRF) of 2100 and 1400 Hz. The corresponding maximum unambiguous velocity for this pulsing scheme 85 is 32.66 m s$^{-1}$. A Doppler clutter filter (Siggia and Passarelli, 2004) was applied and a post-processing algorithm was used to correct spurious Doppler velocities caused by the dual-PRF velocity unfolding (Holleman and Beekhuis, 2003, implementation from https://github.com/meteocat/vcor_dual_prf). The radar data was ingested and processed using the open source package Py-ART (Helmus and Collis, 2016). Other relevant radar specifications are summarised in Table 2.

**Table 2.** Vaisala WRS400 X-band weather radar specifications

| **Radar parameter** | |
| --- | --- |
| Radar Frequency | 9.65 GHz |
| Half Power Beam width | 0.95° |
| Peak power (per channel) | 400 W |
| Pulse length, long | 44 μs |
| Pulse length, short | 1 μs |
| Noise reflectivity @ 1 km | -36.5 dBZ (long pulse) |
| | -21.9 dBZ (short pulse) |
| Pulsing scheme | dual-PRF, 2100/1400 Hz |
| Max. unambiguous velocity | 32.66 m s$^{-1}$ |
| Range resolution | 75 m |
| Max. unambiguous range | 65 km |
| Number of pulses | 64 |
| Elevation angle | 2.0° |



## 2.4 Weather station measurements

The surface station measurements used in the analysis were measured at the Helsinki Airport weather station (WMO code 02974; 60°19'36.1" N 24°57'24.3" E, 51 m above mean sea level) operated by the Finnish Meteorological Institute (FMI). The station is located 6.7 km north-east from the instrument site (see Fig. 2).

Horizontal visibility, taken as a 1-minute average of the measured values, was provided by a Vaisala FS11P present weather sensor. The cloud base altitude was measured by the Vaisala CL31 ceilometer, which has a temporal resolution of 30 s but the

95 cloud base reported by the instrument is an average from several profiles, with the amount of averaging increasing with increasing altitude. A Vaisala FS11P present weather sensor measured the precipitation intensity, taken as the 10-minute average reported every 10 minutes. The measurements are summarised in Table 3.

**Table 3.** Surface weather station measurements used in the analysis.

| Quantity | Aggr. time [min] | Aggregation method | Unit | Resolution | Range | Measurement instrument |
|---|---|---|---|---|---|---|
| Horizontal visibility | 1 | Average | m | 1 | 0...75000 | Vaisala FS11P |
| Cloud base height | 1 | Average | m | 10 | 0...7600 | Vaisala CL31 |
| Precipitation intensity | 10 | Average | $mm\,h^{-1}$ | 0.1 | 0...999.99 | Vaisala FS11P |

When comparing to the measurement times of the lidar and radar measurements, the nearest (in time) reported surface station measurement value was always used. That is, for horizontal visibility and cloud base height, the measurements were taken from

100 the same minute, while for the precipitation intensity the measurement was taken from the nearest complete 10-minute period.

## 3 Methods

Although the Doppler lidar and X-band radar are co-located and observe along the same path, the measurements themselves have different radial and azimuthal resolutions. Therefore, to compare the Doppler velocity measurements, the measurements were interpolated onto a shared Cartesian grid with 250 m × 250 m pixel size. The grid was limited to 14.5 km from the site (i.e.

the maximum range of the lidar). The interpolation was performed with the nearest neighbour interpolation implemented in the wradlib Python package (Heistermann et al., 2013). After interpolation, a median filter was applied in the radar measurements to remove any incorrectly unfolded dual-PRF velocities that were not corrected by the dual-PRF correction algorithm (see Section 2.3).

The pixel-wise agreement of the measurements from closest-in-time lidar and X-band radar scans was then studied using an

110 Ordinary Least Squares regression model. Given the gridded X-band radar measurement $V_i^X$ and lidar measurement $V_i^L$, the regression model is defined as

$$V_i^X = \beta_0 + \beta_1 V_i^L + \epsilon_i,$$ (1)





where $\epsilon_i$ represents the measurement error, assumed to be independent and identically normally distributed, and the coefficients $\beta_0, \beta_1$ are estimated by minimising the residual

$$S(\beta_0, \beta_1) = \sum_{i=1}^{N} \left| V_i^X - \beta_0 - \beta_1 V_i^L \right|^2, \tag{2}$$

where $N$ is the number of paired measurements over the whole measurement campaign.

The goodness-of-fit of the linear regression model can be estimated using the coefficient of determination $R^2$

$$R^2 = 1 - \frac{SS_{\text{res}}}{SS_{\text{tot}}}, \tag{3}$$

where the residual sum of squares $SS_{\text{res}} = S(\beta_0, \beta_1)$, and the total sum of squares

$$SS_{\text{tot}} = \sum_{i=1}^{N} \left( V_i^X - \frac{1}{N} \sum_{i=1}^{N} V_i^X \right)^2. \tag{4}$$

The $R^2$ term indicates the proportion of variance in the data predicted by the model, having values from 0 to 1. The root-mean-squared difference (RMSD) and mean error (ME) were also calculated, with RMSD defined as

$$\text{RMSD} = \sqrt{\frac{1}{N} \sum_{i=1}^{N} \left( V_i^X - V_i^L \right)^2}, \tag{5}$$

and ME as

$$\text{ME} = \frac{1}{N} \sum_{i=1}^{N} V_i^X - V_i^L. \tag{6}$$

The regression model and the statistics were calculated with the statsmodels Python package (Seabold and Perktold, 2010).

Additionally, for each scan, the fraction of measurement bins with valid measurements compared to the total number of bins in the scan was calculated. The fraction was calculated taking into account the blocked beams separately for each instrument, so that a value of 1 indicates that every non-blocked bin had a valid measurement value. These fractions were then used to compare the measurement performance in different weather conditions.

Finally, the performance of the instruments as a function of measurement range was studied by calculating the fraction $f(r)$ of valid measurements at range $r$

$$f(r) = \frac{1}{N_{\text{scans}} N_{\text{rays}}} \sum_{i=1}^{N_{\text{scans}}} \sum_{j=1}^{N_{\text{rays}}} I(r, j, i) \tag{7}$$

where $N_{\text{scans}}$ is the number of scans, $N_{\text{rays}}$ is the number of non-blocked rays for the instrument, and $I(r, j, i)$ is an indicator function for whether the measurement $V_{r,j,i}$ in ray $j$ at range $r$ in scan $i$ is valid:

$$I(r, j, i) = \begin{cases} 1, V_{r,j,i} \text{ valid} \\ 0, V_{r,j,i} \text{ not valid} \end{cases}. \tag{8}$$





## 4 Results

### 4.1 Comparison of measured radial velocities

Figure 3 presents a scatterplot comparing gridded Doppler lidar and X-band radar Doppler velocity measurements. The results
of the statistical comparison are given in Table 4. A value of $R^2 = 0.96$ suggests that the agreement between the two obser-
vations is rather good. We should point out, however, that some artefacts can be seen in Fig. 3. For example, there are some
points where the lidar is measuring a zero velocity, but the X-band radar is not. This is due to differences in the processing and
clutter filtering.

The statistics in Table 4 show that the mean error between the measurements is only $-0.047\,\mathrm{m\,s^{-1}}$. Additionally, there is
145 no significant correlation between the linear regression residual and lidar CNR or X-band SNR (not shown). Therefore, we can
see that the measurements by the two instruments agree well, and possible differences in the measurements are better explained
by the different measurement volumes and scatterers, and the time difference between the measurements, than instrument error
or bias between the instruments.

**Table 4.** Statistics on the agreement of Doppler lidar and X-band radar radial Doppler velocity measurements.

| Statistic | Value |
|---|---|
| N | 11 198 981 |
| RMSD | $1.31\,\mathrm{m\,s^{-1}}$ |
| ME | $-0.047\,\mathrm{m\,s^{-1}}$ |
| $R^2$ | 0.96 |
| Slope | 1.017 |
| Intercept | $-0.040\,\mathrm{m\,s^{-1}}$ |

Note that Fig. 3 and the statistics in Table 4 have been calculated from measurements where both Doppler lidar and X-band
radar measurements were available, which in our data corresponds to approximately 5% of all measurements. The differences
in measurement availability for the two instruments are discussed in the following sections.

### 4.2 Measurement performance

The data availability, computed as the fraction of measurements available at each range bin over the entire measurement
campaign period, provides a description of the overall performance of each instrument. This is shown in Fig. 4 as the fraction of
155 measurements available with range over the entire measurement campaign period. The availability as function of range depends
on both the specific instrument and local conditions, e.g. aerosol content for a Doppler lidar, therefore, when examining the
measurement performance in different conditions, we need to consider the effect relative to Fig. 4.

The Doppler lidar displays very high availability at short ranges, with availability decreasing almost linearly with range. At
14.3 km in range, the measurement availability is about 12%.



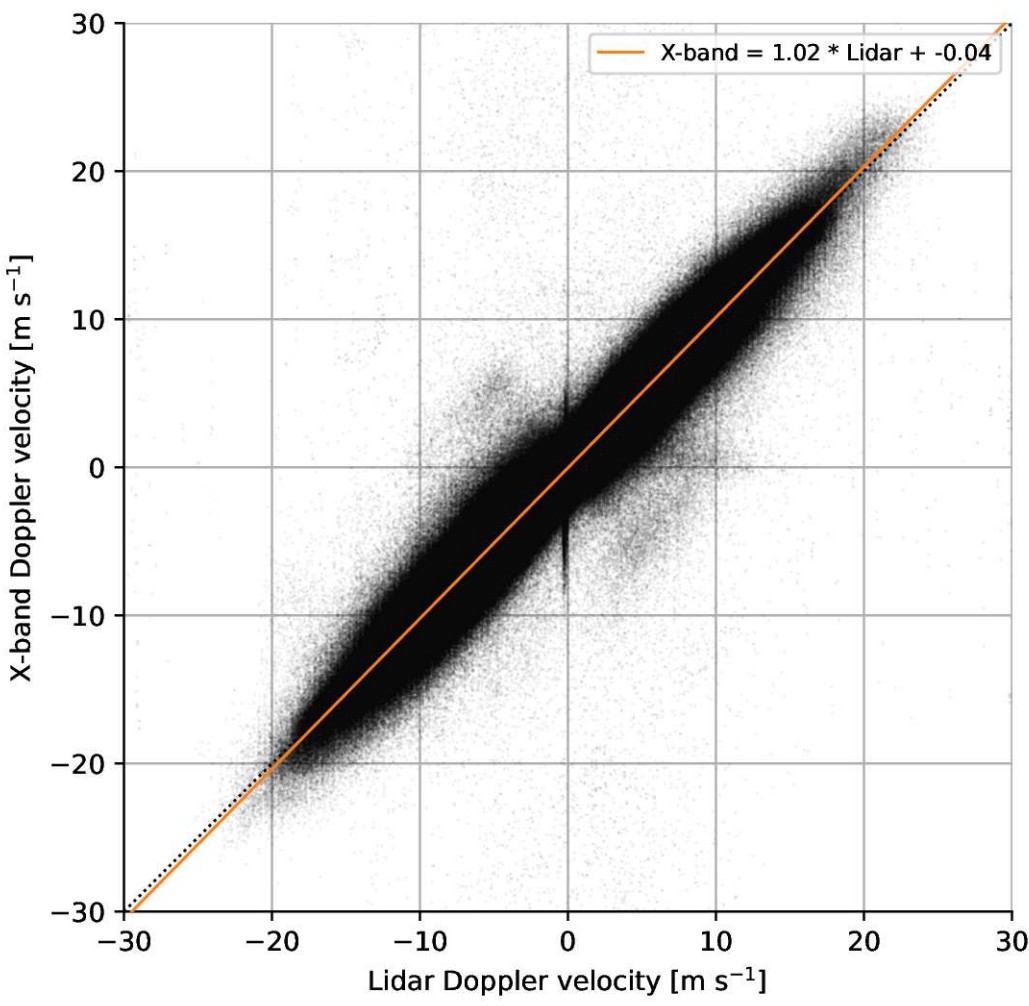

**Figure 3.** Scatterplot of Doppler lidar versus X-band radar radial Doppler velocity measurements. The orange line indicates the linear fit to the data and the black dashed line indicates the one-to-one agreement.





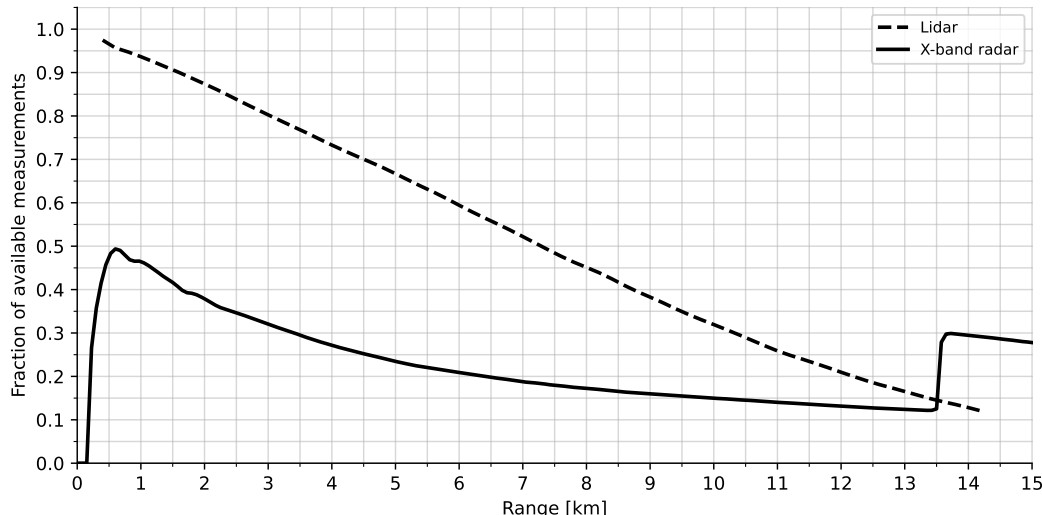

**Figure 4.** Fraction of available measurements, i.e. data availability, as a function of measurement range for Doppler lidar (dashed line) and X-band radar (solid line) measurements.

For the X-band radar, the overall availability is much lower than for the Doppler lidar. Again, availability is better at short ranges and decreases with range. The change in pulse length at 13.5 km is clearly visible, with the 14.6 $\mathrm{dB}$ greater sensitivity of the long pulse responsible for the increase (see Table 2). Note that here, there has been no blending of short and long pulse.

### 4.3 Measurement performance as function of horizontal visibility

Next we study the effect of horizontal visibility on the measurement performance. Figure 5 presents the distribution of the fraction of valid measurements for both instruments as a function of horizontal visibility, separated into summer (May to September) and autumn (October to November) months. Only measurements with a corresponding valid horizontal visibility measurement were included, which was approximately 90% of all measurements. The Doppler lidar measurements exhibit similar distributions, even though the majority of the cases in the summer months have visibility above 35 km (Fig. 5a). Figures 5a and 5c show a linear increase in Doppler lidar measurement availability with horizontal visibility up until a maximum at about 40 km after which the availability then decreases, presumably due to the decreasing atmospheric aerosol number concentrations responsible for increasing visibility.

A very different response is seen for the X-band radar. In the summer months (Fig. 5b), insects are responsible for a large portion of the distribution towards the lower right quadrant (fraction values from 0 to 0.9 and horizontal visibility above 30 km), which then mostly disappear during the autumn (Fig. 5d). The portion of high fractions with low visibility in the top left corner are attributed to precipitation and are present during both seasons.

These effects can also be seen in Fig. 6, which shows the measurement availability as a function of range for different horizontal visibility intervals. For low horizontal visibilities, the Doppler lidar measurement availability decreases rapidly

**Figure 5.** Fraction of valid measurements as function of horizontal visibility from May to September and October to November 2021 for (a, c) Doppler lidar and (b, d) X-band radar



with increasing range (Fig. 6a). As visibility increases, the Doppler lidar measurement availability increases up until visibility

reaches about 40 to 50 km, after which the availability decreases. The relatively larger decreases seen beyond a distance in

range of 4 km for some curves is likely due to the shape of the telescope focus function that the Doppler lidars are configured

with (Pentikäinen et al., 2020).

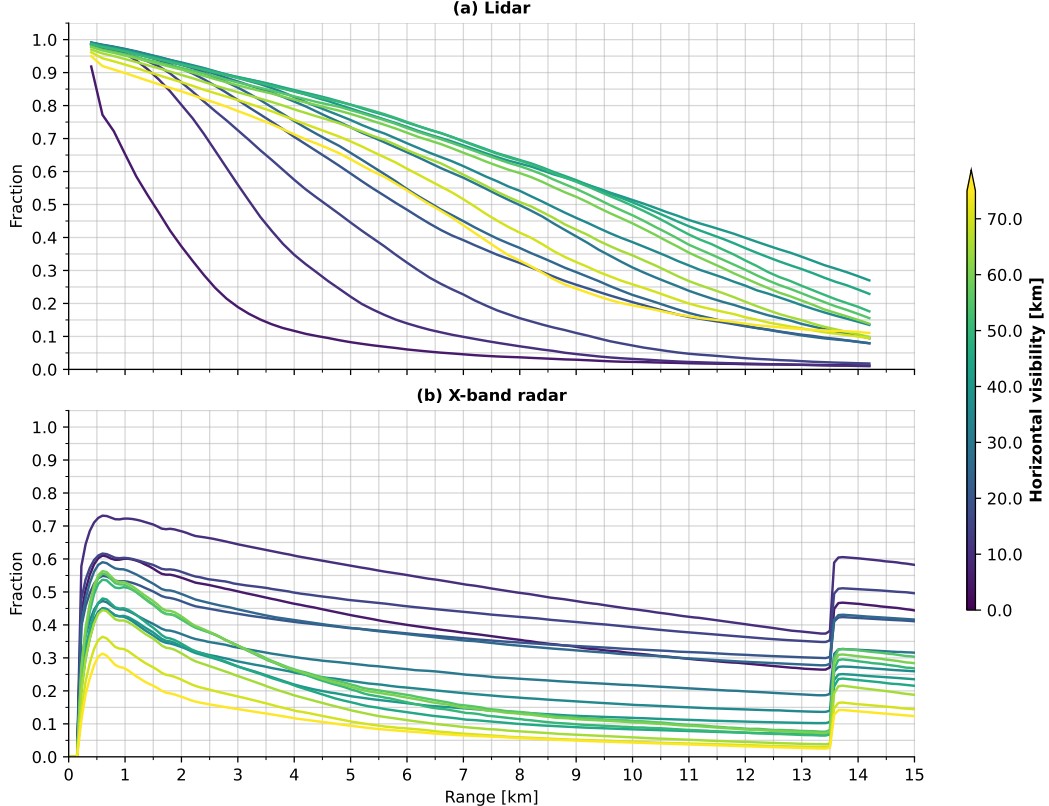

**Figure 6.** Fraction of available measurements as a function of measurement range for (a) Doppler lidar and (b) X-band radar calculated

separately for different horizontal visibility classes. Each line corresponds to a horizontal visibility class with a bin width of 5000 m.

For the X-band radar (Fig. 6b), the highest measurement availability at all ranges is achieved at low horizontal visibility

due to precipitation. Note that since the low horizontal visibility cases will also include situations with fog where the X-band

radar will not measure well, the average measurement availability of the cases here is at most 0.7. The availability decreases

as the horizontal visibility increases, and the lowest availability is obtained at highest horizontal visibility values. The jump in

availability at the pulse change is clearly visible, but the horizontal visibility affects the magnitude of the jump, with the largest

jump occurring at low visibility.

Two example cases demonstrating the discussed effects are shown in Figures 7 and 8. Both cases are clear air cases with no

precipitation. During the first case in Fig. 7 from August 28th, 2021, the X-band radar had a strong insect echo and thus was





able to measure throughout almost the entire measurement area (excluding beam blockage). The horizontal visibility during
the measurements was approximately 45 km, resulting in the Doppler lidar being able to measure out to close to the maximum
range.

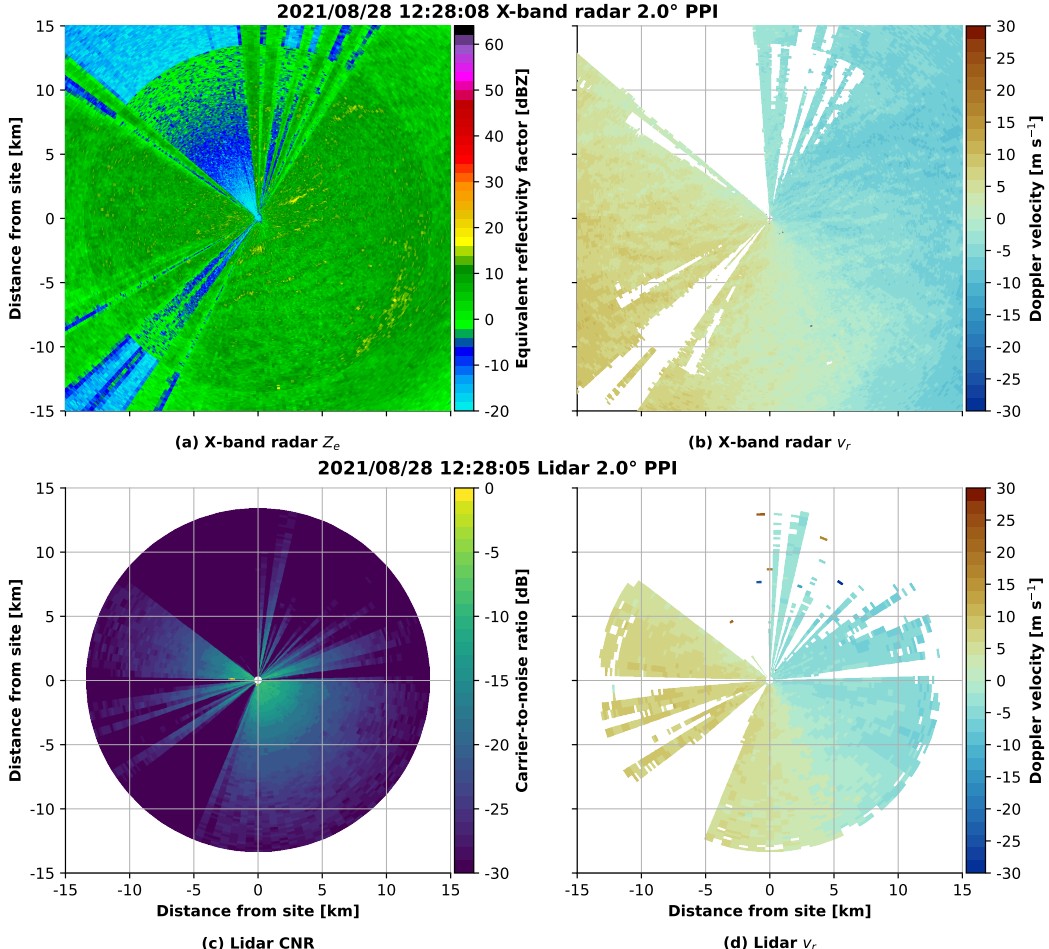

**Figure 7.** Data example from August 28th 2021 at 12:28 UTC showing PPI scans at 2° of X-band (a) radar reflectivity and (b) Doppler velocity, and Doppler lidar (c) carrier-to-noise ratio (CNR) and (d) Doppler velocity. Doppler velocity colour map is from Crameri (2021).

The second case (Fig. 8) on June 17th, 2021, on the other hand, occurred on a day with few insects in the boundary layer, so the X-band radar had no measurements beyond the first few kilometres. The horizontal visibility during the case was
approximately 63 km, and correspondingly the Doppler lidar was not able to measure consistently beyond 10 km in range.



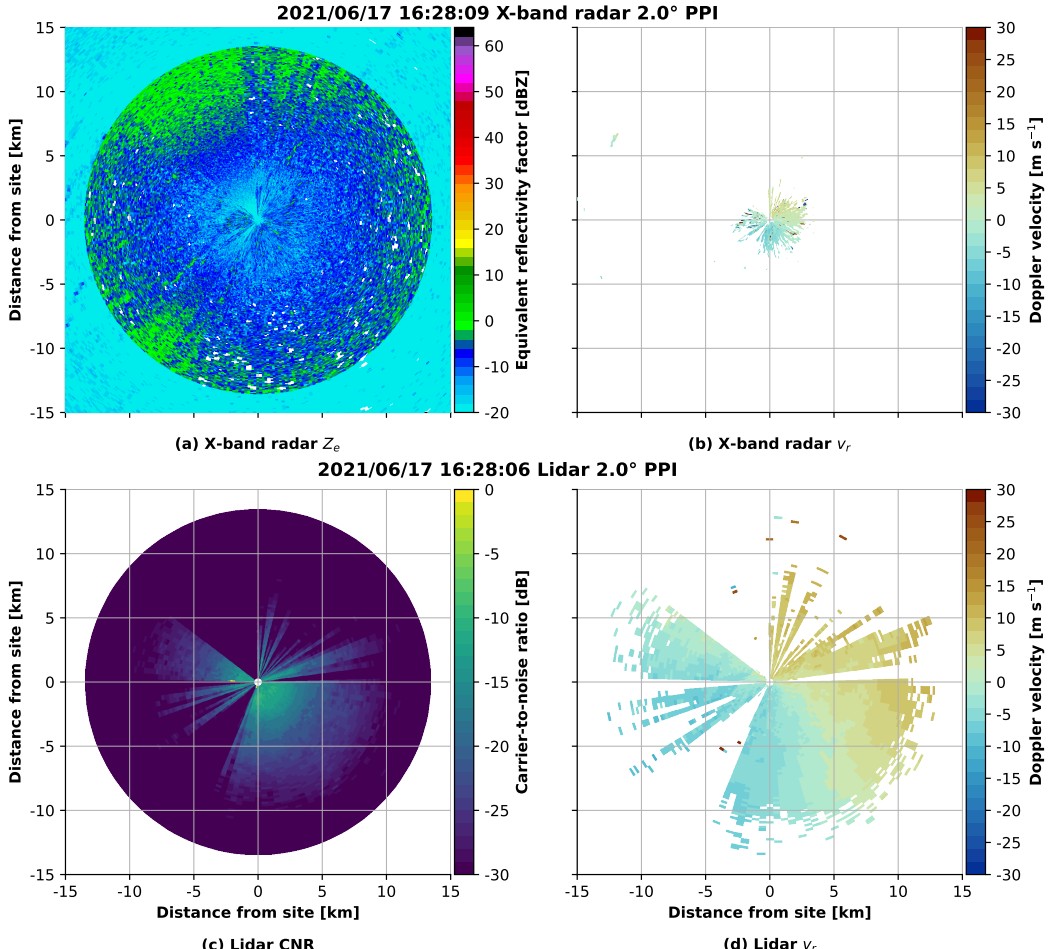

**Figure 8.** Data example from June 17th 2021 at 16:28 UTC showing PPI scans at 2° of X-band (a) radar reflectivity and (b) Doppler velocity, and Doppler lidar (c) carrier-to-noise ratio (CNR) and (d) Doppler velocity. Doppler velocity colour map is from Crameri (2021).

### 4.4 Measurement performance as function of cloud base height

Similar analysis as for the horizontal visibility was performed with respect to cloud base height. The cloud base height analysis was limited to the lowest 3 km, as this was considered most relevant for scanning at low elevation angles. Only measurements with a corresponding valid cloud base height measurement were included, which was approximately 72% of all measurements.

Figure 9 shows how the fraction of valid measurements varies with respect to cloud base height. The distributions show the impact of the severe in-cloud attenuation experienced by Doppler lidar. Assuming a reasonably homogeneous cloud layer, the valid Doppler lidar measurements are limited to the range at which the lidar beam impinges the cloud layer. A cloud layer at 250 m would be intercepted at about 7 km in range by a lidar beam scanning at 2 degrees in elevation, implying a maximum availability of 0.5 since the maximum range is about double this; this indeed corresponds to the value we observe for a cloud





base height of 250 m. This is the case for both summer and autumn. When the cloud layers are at higher altitudes, and not intercepted by the lidar beam, summer months still show reasonable availability, which is not dependent on cloud base height. It is not certain whether the different response seen in cloud base height above 1 km in autumn months is due to a lack of clouds at these heights in our dataset.

The X-band radar exhibits a more bimodal distribution with respect to cloud base height (Fig. 9b, d). For all cloud base heights shown, the availability of radar measurements is either mostly low, which we attribute to non-precipitating clouds, or mostly high, which we attribute to precipitating clouds. Cloudy cases with intermittent precipitation are presumably responsible for the low cloud situations with valid measurement fractions between 0.1 and 0.9, and these appear to be more common in winter months. The increase in scatter observed during the summer months (Fig. 9b) for cloud bases at higher altitudes is likely due to the presence of insects in the boundary layer below clouds, especially likely below non-precipitating clouds.

Figure 10 shows the measurement availability as function of range and cloud base height. For Doppler lidar, below 500 m in cloud base height, the availability is very dependent on cloud base height as shown previously. Once the cloud base height is above 500 m, the lidar beam will no longer be impacted by the cloud itself and there is a less rapid increase in availability with cloud base height. This suggests that there is still some reduction in availability due to attenuation by precipitation falling from these clouds, the amount of which reduces as the cloud base height increases further.

The performance of the X-band radar again shows different behaviour to the Doppler lidar. For the X-band radar, low cloud base height results in high availability, with the availability reducing as the cloud base height increases, especially for cloud base heights above 1 km. As for Doppler lidar, this can be attributed to the same reason, the amount of precipitation falling into the radar beam reduces as the cloud base height increases, except that now this reduces data availability rather than increases. The relative reduction in availability with range is more rapid for high cloud base heights than for low cloud base heights.

## 4.5    Measurement performance as function of precipitation intensity

Figure 11 shows the distributions of fraction of valid measurements as function of precipitation intensity for the Doppler lidar and X-band radar. Only measurements with a corresponding valid precipitation intensity measurement were included, which was approximately 56% of all measurements. For Doppler lidar (Fig. 11a), the presence of precipitation reduces measurement availability due to attenuation of the lidar beam. However, precipitation is also a strong scatterer at lidar wavelengths and attenuation is not as rapid as for liquid cloud, hence the Doppler lidar is still capable of measuring up to a kilometre or more into precipitation, and further if the precipitation is patchy. For the X-band radar (Fig. 11b), as can be expected, the presence of any precipitation will improve availability.

Figure 12 shows the effect of precipitation intensity on the measurement availability as function of range. For Doppler lidar (Fig. 12a), increasing precipitation intensity decreases the availability, but this effect is only evident at ranges beyond 3 km.

The X-band radar has high availability in precipitation (Fig. 12b) with slight reductions in availability for precipitation intensities less than 0.5 mm h$^{-1}$. Note that even for high precipitation intensities, the increase in availability for the longer pulse length is still evident.

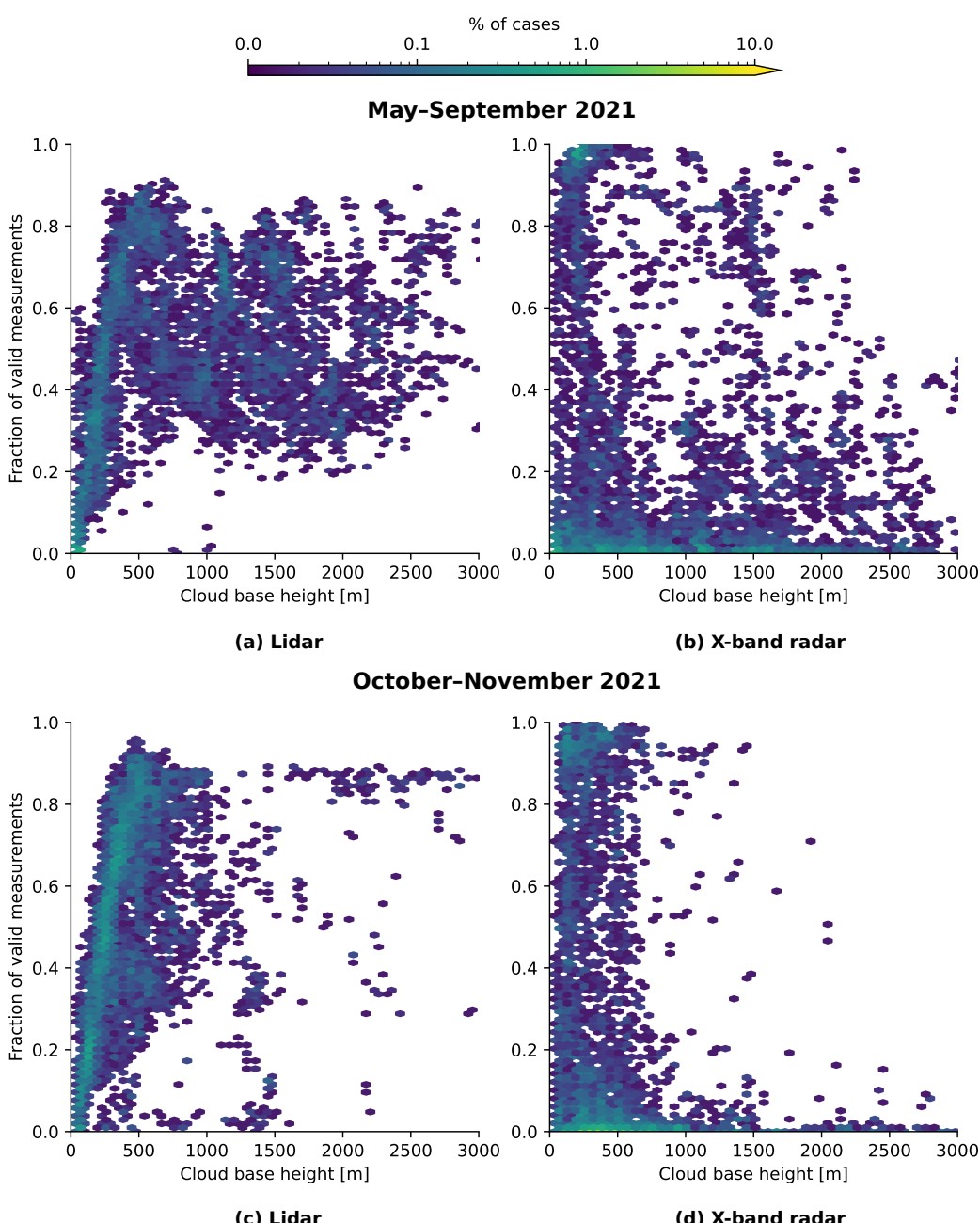

**Figure 9.** Fraction of valid measurements as function of cloud base height from May to September and October to November 2021 for (a, c) lidar and (b, d) X-band radar.

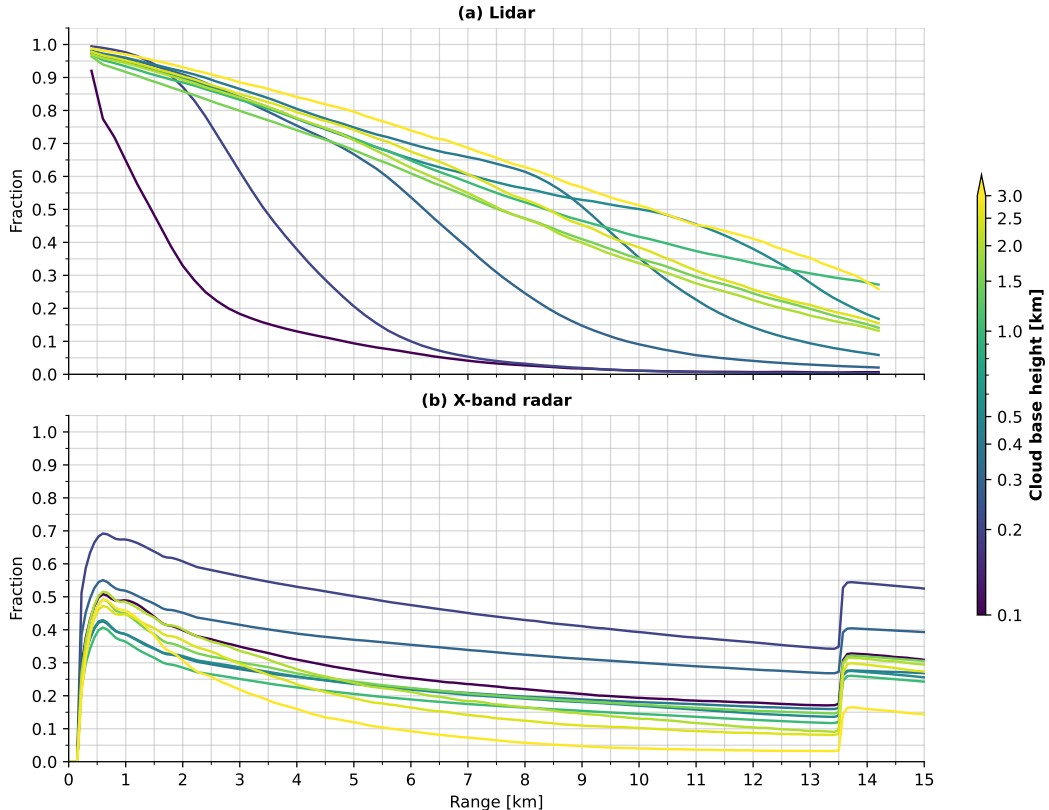

**Figure 10.** Same as Fig. 6, but as a function of cloud base height, separated into different cloud base height classes. Each line corresponds to a cloud base height class with a bin width of 100 m below 500 m and a bin width of 500 m above. The line colour indicates the upper limit of the bin. Note the logarithmic scale for the colour bar.

Figure 13 shows an example of the Doppler lidar and X-band weather radar measurements from a precipitation case from May 17th, 2021. During this time, a thunderstorm was passing over the measurement area. The X-band radar provided excellent

measurement coverage inside the precipitation in the eastern half of the area, but hardly any measurements in the clear air area in the west. On the contrary, the Doppler wind lidar obtained measurements in the clear air area, but the signal was attenuated quickly inside the heavy rainfall.

## 5 Discussion and conclusions

The aim of this study was to compare Doppler velocities measured by Doppler lidar and X-band weather radar, to see, firstly,

how well the co-located measurements agree and, secondly, how the availability of the measurements varies in different weather conditions. When Doppler velocity measurements are available from both Doppler lidar and X-band weather radar, the agree-



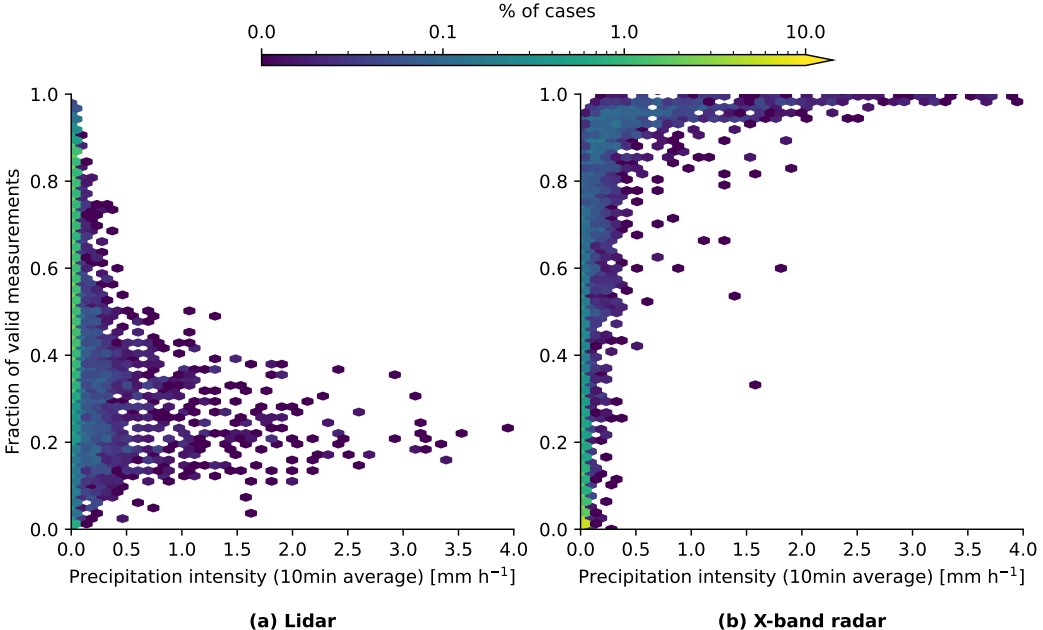

**(a) Lidar**     **(b) X-band radar**

**Figure 11.** Fraction of valid measurements as function of precipitation intensity (10-minute average, optical) for (a) Doppler lidar and (b) X-band radar Doppler velocity measurements.

ment is good with $R^2 = 0.96$. Some departures in agreement were apparent for specific Doppler velocities, indicating that, in some situations, it might be necessary to apply clutter filtering to Doppler lidar measurements at low elevation angles.

The two instruments operate at very different wavelengths and have very different responses to the scattering media present in the atmosphere. Therefore, as expected, the two instruments exhibited different data availabilities. If their different capabilities in specific weather situations can complement one another, then their combination would increase the spatial coverage of wind observations across a wide range of weather conditions.

To better understand the response of each instrument to different meteorological conditions, we investigated their measurement availability in terms of horizontal visibility, cloud base height, and precipitation intensity. In general, Doppler lidar and X-band radar displayed contrasting measurement availabilities in the meteorological conditions examined here. Precipitation, often accompanied by low horizontal visibility and low cloud base height, lead to high radar measurement availability, whereas the Doppler lidar measurement availability was reduced due to the signal being attenuated. Our study showed that attenuation of the Doppler lidar signal by precipitation was significant but not severe, with reasonably good measurement availability out to at least 2 km, which is in agreement with Nijhuis et al. (2018). Attenuation by cloud however, is very rapid, and the Doppler lidar does not measure much more than 300 m beyond cloud base.

In contrast, in clear air situations, with high horizontal visibility and high cloud bases, the Doppler lidar is able to measure out to longer ranges. The only hindrance for lidar measurements in clear air is when the atmospheric aerosol concentration falls





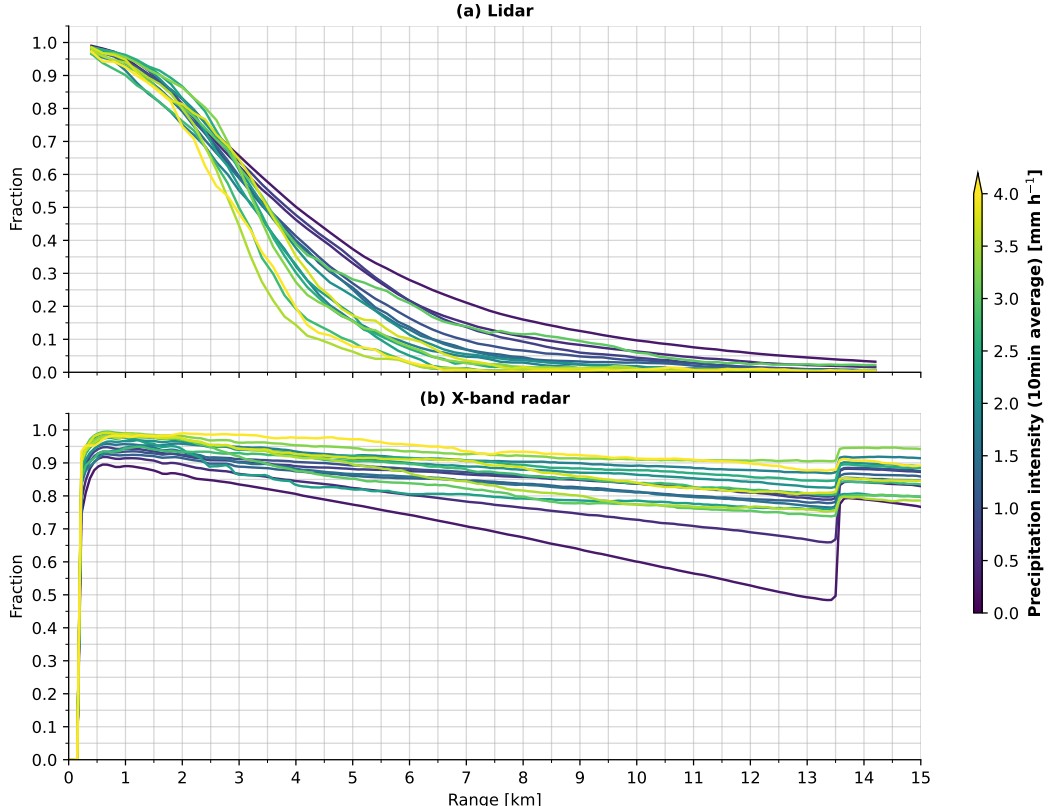

**Figure 12.** Same as Fig. 6, but as a function of precipitation intensity, separated into different precipitation classes. Each line corresponds to a precipitation intensity class with a bin width of $0.25 \, \mathrm{mm \, h^{-1}}$

.

so low that there is no longer sufficient signal, as in cases with extremely high horizontal visibility. For the radar, there may not be enough scatterers for the radar to measure consistently in clear air. One exception to this are insects which can increase

the radar measurement availability substantially in daytime during summer. This leads to a seasonality in the availability of radar clear air measurements, seen in Fig. 5, where in summer the availability can be quite high, but in other seasons no measurements are obtained.

Our measurements show that the conditions where both Doppler lidar and the X-band radar are able to obtain Doppler velocity measurements are the summer clear air situations where insects and aerosol are present. During our measurement

campaign, the optimal horizontal visibility in these cases was approximately $40 \ldots 60 \, \mathrm{km}$ and the cloud base height higher than $1 \, \mathrm{km}$. However, optimal conditions depend on the measurement location, as for example horizontal visibility can be decreased by poor air quality, which will also attenuate the Doppler lidar signal more.

It is also of interest is to determine whether there are conditions where neither instrument is able to measure Doppler velocity and provide winds. For example, conditions with low cloud base height but with no precipitation, including fog, can

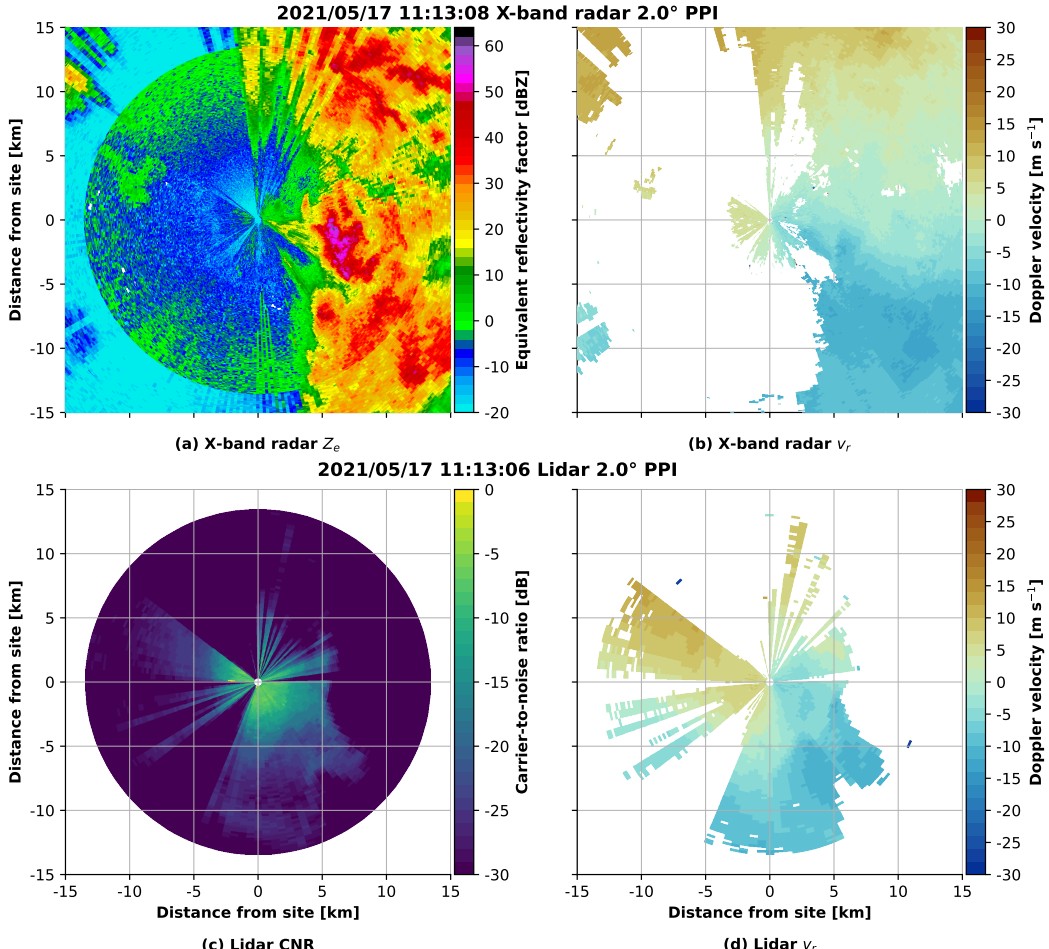

**Figure 13.** Data example from May 17th 2021 at 11:13 UTC showing PPI scans at 2° of X-band (a) radar reflectivity and (b) Doppler velocity, and Doppler lidar (c) carrier-to-noise ratio (CNR) and (d) Doppler velocity. Doppler velocity colour map is from Crameri (2021).

be responsible for poor data availability from both instruments, especially for measurements at long distances. However, our analysis did not reveal a significant proportion of situations where measurements from both instruments were not available. As the data availability for radar and lidar exhibits contrasting behaviour, it is likely that at least one of the instruments will provide Doppler velocity observations.

*Code and data availability.*   The code used for the analysis is available at https://github.com/ritvje/lidar-xband-article-scripts. Numerical ver-
280 sions for the data availability curves in Figures 4, 6, 10 and 12 can be found at http://doi.org/10.23728/fmi-b2share.9e5071ae63dd42ad9ca64016796ec817. The Doppler lidar and X-band radar data are available upon request from the corresponding author. The observations from the FMI automatic weather station used in this study are available at the FMI open data portal (https://en.ilmatieteenlaitos.fi/open-data).



*Author contributions.* Conceptualization: JR, EOC, DM, RL, JT, LT. Data curation: JR, RL. Analysis: JR. Funding acquisition: JR, RL, JT. Investigation: JR. Methodology: JR. Project administration: RL, JT. Resources: RL, JT, LT. Software: JR. Supervision: EOC, DM, RL, JT,

LT. Validation: JR. Visualization: JR, RL. Writing - original draft: JR, EOC, DM. Writing - review & editing: JR, EOC, DM.

*Competing interests.* JR, EOC, DM and JT have no competing interests. RL and LT are employees of Vaisala Oyj.

*Acknowledgements.* This work was funded through the MWS-A project funded by the European Space Agency (4000132768/20/UK/ND).





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
