# Peer review of "Complementarity of Wind Measurements from Co-located X-band Weather Radar and Doppler Lidar"

_Atmospheric Measurement Techniques, 2022_

## Author Response (AR1)

**Response to Reviewers – "Complementarity of Wind Measurements from Co-located X-band Weather Radar and Doppler Lidar"**

Jenna Ritvanen, Ewan O'Connor, Dmitri Moisseev, Raisa Lehtinen, Jani Tyynelä, and Ludovic Thobois

October 18, 2022

We thank the two reviewers and Sebastian Kauczok for their constructive comments on our submitted manuscript. We have copied the comments of the reviewers here and reply to each comment below it.

Please note that in addition to changes to address the comments, detailed in the replies, the following changes were made to the manuscript:

1. Due to the analysis added to discuss insects as scatterers, also the analysis in Section 4.1 was re-run over a slightly larger data set. As a result, the values presented in Table 4 and mentioned in the section have changed and Figure 3 was updated. The changes in the values occurred in mostly in the second or third decimal place and they do not change the conclusions in the manuscript.

2. The colormap for radar reflectivity in Figures 7, 8 and 13 (14 in updated version) was changed to be more colorblind-friendly.

**Reviewer 1**

The number of pulses in Table 1 and Table 2 is confusing to me. E.g. for the lidar: It is written that the number of pulses is 1000. What is meant by "the number of pulses"? Normally I would say that 1000 pulses are used to calculate the velocities in a 3.0 degree ray. With 1000 pulses and a PRF of 10 kHz, an accumulation time of 0.1 s and an azimuth rate of 30 deg/second is used. I can't imagine that with an accumulation time of 0.1s, the data availability would be as good as reported in this article.

Please clarify what the measurement specifications are.

**Reply to review comment 1**

We thank the reviewer for catching the error in the measurement specifications. In Table 2, the number of pulses in the lidar measurements was updated to be 10 000. The accumulation time of the measurements is 1000 ms, with each PPI scan lasting approximately 2 minutes.

We also checked the values in Table 1 for the radar measurement specifications and found no errors there.

**Reviewer 2**

Novel ideas concerning accurate wind measurements are presented in this manuscript. Accurate measurements of wind profiles are of the essence given the use of these measurements, for example, in numeric weather prediction models. The results presented in the manuscript are based on data (i.e., Doppler velocities) collected from two co- located instruments: an X-band weather radar and a Doppler lidar. The data, collected during a 7-month campaign conducted in Vantaa (Finland) show that the conditions in which both instruments were able to provide Doppler velocities measurements are the summer clear air situations in which both insects and aerosols are present. Thus, The data from these two instruments can be complementary in certain weather situations and using them in synergy can improve the spatial coverage of wind measurements for different weather situations.

The subject of the manuscript is within the scope of AMT. The manuscript is very well written, clearly outlined and with a complete description of the methods to allow reproduction.

One minor comment concerning the use of hexagons to represent data points in Figures 5, 9, and 11. Are the hexagons used to improve spatial representation (compared with squares)? Furthermore, the hexagons are stretched in the horizontal direction in Figures 9 and 11.

**Reply to review comment 2**

We thank the reviewer for the kind comments and apt summary of the manuscript.

Regarding Figures 5, 9 and 11, the histograms were plotted using hexagons (matplotlib hexbin-function) instead of rectangles (hist2d-function) to improve visual representation. The hexagons in Figures 9 and 11 are stretched in the horizontal due to the different bin sizes. The horizontal bin size in Figure 5 for horizontal visibility is 2500 meters, in Figure 9 for cloud base height 50 meters, and in Figure 11 for precipitation intensity 0.015 $\text{mm}\,\text{h}^{-1}$. The bin sizes were selected by hand to achieve a reasonable balance between the resolution in the histogram and the number of bins, as using too many bins would have led to very low values (percentage of cases) in each bin, which again would have made the figures harder to interpret. Additionally, each figure was plotted to have approximately equal-sized x- and y-axes. As a result, in Figures 9 and 11 the hexagons are slightly stretched in horizontal, while in Figure 5 the hexagons appear symmetric.

**Community comment from Sebastian Kauczok: Bias in Doppler Velocity due to Insect Migration**

Dear authors,

Throughout your contribution you treat insects as proper and unproblematic targets for Doppler weather radar wind velocity measurements. There is neither a discussion of the potential bias in velocity due to migratory behaviour of insects, nor a verification that this is not the case in your data set. It is known in the literature that special heed needs to be payed, if insects are acceptable as a target or not for the application at hand. In the latter case, insects need to be filtered out (see [1] and [2], for example). For example, the German Weather Service operates X-Band Doppler weather radar/Doppler Lidar combinations at Frankfurt and Munich airport. Their observations reveal that differences between Doppler velocities from radar and lidar are unacceptable, if insects are not filtered out, since this leads, inter alia, to false wind shear alerts. [3]

[1] Rennie, S.J.: "Doppler weather radar in Australia.", CAWCR Technical Report No. 055, 2012

[2] Hannesen, R., S. Kauczok, and A. Weipert: "Quality of clear-air radar radial velocity data: Do insects matter?" 8th European Conference on Radar in Meteorology and Hydrology, Garmisch-Partenkirchen, Germany. 2014

[3] B. Stiller, German Weather Service, personal communication, 2013

**Reply to community comment from Sebastian Kauczok**

We thank the author for the comment and pointing out a considerable shortcoming in the analysis presented in the manuscript. To investigate the issue whether insects are biased scatterers for the radar, we added a scatterplot (Fig. 9 in manuscript) to Section 4.3 that contains only measurements where the X-band radar has $Z_{DR} \geq 5$ and $\rho_{HV} \leq 0.9$, as those measurements would be expected to be from insects. For this subset of the data, the coefficient of determination $R^2 = 0.95$ is slightly decreased and bias (ME $= 0.078 \, \text{m s}^{-1}$) increased, but the RMSD $= 1.13 \, \text{m s}^{-1}$ is decreased. Visually, the artefacts seen in Fig. 3 are not present in the new scatterplot. This leads us to conclude that for our location and data, insects are not an issue when comparing Doppler velocity measurements from lidar and radar. However, this is certainly not the case everywhere, so the issue should be investigated separately for each location.

We have added discussion summarizing the above in Section 4.3 and in the conclusions in Section 5. Additionally, we added discussion of dual-PRF unfolding errors as an error source in Section 4.1.

[Figure]

Figure 1: Scatterplot of Doppler lidar versus X-band radar radial Doppler velocity measurements where X-band radar $Z_{\mathrm{DR}} \geq 5$ and $\rho_{\mathrm{HV}} \leq 0.9$. The orange line indicates the linear fit to the data and the black dashed line indicates the one-to-one agreement.